# ShowMaker: Creating High-Fidelity 2D Human Video via Fine-Grained Diffusion Modeling

Quanwei Yang [1†]    Jiazhi Guan [2]    Kaisiyuan Wang [3*]    Lingyun Yu [1]
Wenqing Chu [3]    Hang Zhou [3]    Zhiqiang Feng [3]    Haocheng Feng [3]
Errui Ding [3]    Jingdong Wang [3]    Hongtao Xie [1*]

[1] University of Science and Technology of China    [2] Tsinghua University
[3] Department of Computer Vision Technology (VIS), Baidu Inc.

## Abstract

Although significant progress has been made in human video generation, most previous studies focus on either human facial animation or full-body animation, which cannot be directly applied to produce realistic conversational human videos with frequent hand gestures and various facial movements simultaneously. To address these limitations, we propose a 2D human video generation framework, named *ShowMaker*, capable of generating high-fidelity half-body conversational videos based on 2D key points via fine-grained diffusion modeling. We leverage dual-stream diffusion models as the backbone of our framework and carefully design two novel components for crucial local regions (i.e., hands and face) that can be easily integrated into our backbone. Specifically, to handle the challenging hand generation caused by sparse motion guidance, we propose a novel Key Point-based Fine-grained Hand Modeling module by amplifying positional information from raw hand key points and constructing a corresponding key point-based codebook. Moreover, to restore richer facial details in generated results, we introduce a Face Recapture module, which extracts facial texture features and global identity features from the aligned human face and integrates them into the diffusion process for face enhancement. Extensive quantitative and qualitative experiments demonstrate the superior visual quality and temporal consistency of our method.[1]

## 1   Introduction

The recent advancement of generative models [5, 21, 11, 36, 33] has significantly propelled digital human technology, which is widely applied in business, education, and multimedia entertainment. Empowered by these generative models, numerous works [30, 40, 3, 52, 14, 32, 4, 24, 15, 48, 6, 16, 18, 34, 23, 35, 54, 49] have given their priority to 2D video synthesis to human generation. Though most of them focus on the head region, few studies [16, 18, 22, 34, 23, 35, 54, 37, 45, 25] attempt to generate full-body videos by animating a reference image with a sequence of driving motion signals via a warping paradigm, but their results fall short in terms of both generation quality and temporal consistency. However, the demand for creating high-fidelity 2D avatars under more challenging conversational scenarios (e.g., TV shows and stand-up comedy) is increasing rapidly, which cannot be fulfilled with such a fidelity state.

Recently, efforts have been made to investigate architectures built upon pre-trained diffusion models for controllable human body animation [1, 13, 47]. These methods adopt a dual-stream design to

---

† Work done during an internship at Baidu Inc.
* Correspondence to: `htxie@ustc.edu.cn`; `wangkaisiyuan@baidu.com`
[1] Project webpage: https://mumuwei.github.io/ShowMaker

separately model the textural information from the reference image and the motion information from dynamic 2D skeletons [50] or 3D reconstructions [28, 55]. Effective interaction is also achieved between the two streams via widely used cross-attention [39]. Despite their captivating performance in full-body animation, we identify three major challenges that require rethinking. a) Creating a human-like conversational avatar requires a holistic synthesis of the human body. Though concurrent studies [17, 56] leverage 3D body reconstruction, which contains rich depth and shape information in face and hand regions, it suffers from occasional incorrect pose estimation for hands or temporal jittering problems. In addition, tedious pre-processing and time-consuming per-frame optimization [51] are required, which cannot be applied in a user-friendly system. We find using 2D representations more stable and efficient. b) It is quite intricate to generate human hands with sparse representations. Human hand regions occupy only a limited number of pixels in the original video frame. This easily leads to unstable hand synthesis with blurry texture or incorrect shape. Naive designs in [1, 13] cannot achieve detailed texture synthesis and delicate control in such local regions (e.g., faces and hands). c) Facial identity preservation is another problem that has not been well investigated. It becomes particularly challenging for these previous works [1, 13] due to the strong entanglement between identity information and 2D driving signals, which inevitably leads to identity degradation during cross-identity animation. The recent study [17] attempts to involve more controllable signals (i.e., 3D morphing parameters) for face enhancement, but the identity preservation performance is still not satisfying enough.

To tackle the above challenges, in this paper, we propose a holistic human video generation framework named *ShowMaker*, capable of generating an expressive conversational human video with fine-grained modeling conditioned solely on 2D key points. We adopt a similar dual-stream architecture as in [1, 13, 47, 56] as the backbone of our framework and introduce two new designs. We first introduce a novel *Key Point-based Fine-grained Hand Modeling* module to reliably recover hand regions in detail from sparse driving guidance. Our key insight is to leverage resolution-independent representations (i.e., the coordinates of the hand key points) to provide stable structure guidance. In terms of hand texture, we novelly design a key point-based hand texture codebook equipped with a set of learnable basis vectors, which has a one-to-one correspondence with the hand key point topology. Specifically, we predict a set of weights from the hand key points of each hand and produce key point-based hand textural compensation through the weighted combination, which can be directly injected into the diffusion process via cross-attention. Furthermore, to improve the facial region quality of the target subject, we propose a *Face Recapture* module to construct comprehensive face representations by performing structure encoding, texture encoding, and identity encoding simultaneously. This multi-level encoding strategy can significantly alleviate the identity leakage issue during cross-identity animation on 2D avatars, which is not available in other comparative methods. Extensive quantitative and qualitative experiments demonstrate that our proposed method can synthesize 2D human video with better visual quality and more accurate body movements, especially hand gestures.

The main contributions of this paper are summarized as follows: (1) We propose a novel holistic human video generation framework with fine-grained modeling, named *ShowMaker* for creating 2D human conversational videos conditioning on 2D key points. (2) We propose a *Key Points-based Fine-grained Hand Modeling* module, which achieves robust hand synthesis via a key point-based codebook. (3) We propose a *Face Recapture* module, which can effectively recover richer facial details and recapture the identity of the target subject.

## 2   Related Work

**Talking Head Generation.** The goal of talking head synthesis is to generate realistic face videos based on a driving video or audio. Depending on the driving signal, talking head synthesis can be divided into video-driven methods and audio-driven methods. Video-driven methods [43, 53, 3, 32, 4, 48, 42] aim to establish the mapping between the reference face and the driving head motion, such as head/mouth movements, expressions, etc. For instance, DaGAN [12] learns to face depth maps and 3D facial geometry in a self-supervised manner to estimate motion fields. While MetaPortrait [53] estimates the warping flow between the driving and reference face through predefined dense facial key points. Audio-driven methods[40, 52, 41, 7, 24, 15, 6, 46] focus on ensuring the generated mouth movements are synchronized with the given audio. For example, Wav2lip [30] edits the mouth area of the face video based on the input audio signal and employs a discriminator to ensure lip-sync. Similarly, StyleSync [6] generates lip-sync face videos by combining audio information with masked

facial spatial information in style space. Although these methods can produce high-quality face videos, they lack comprehensive pose movements and gestures, limiting their application scenarios.

**Human Body Animation** The human video synthesis task aims to generate a full-body human video with the reference person appearance and driving pose. According to the pipeline, these methods can be divided into two categories: implicit methods and explicit warp-based methods. For the first category, some early GAN-based algorithms [16, 18, 37] map or manipulate the reference appearance to the driving pose in the latent space, subsequently generating the target video. In recent years, significant progress has been made in human video generation by leveraging powerful diffusion models. Some approaches [44, 19, 47]integrate the reference appearance details with driving information through cross-attention mechanisms in the Denoising U-Net, resulting in enhanced generation quality. The warp-based methods [34, 23, 35, 54, 25, 45] warp the reference image or its features to the driving pose by various estimated flows, such as 2D optical flows, and 3D flow fields. In addition to the aforementioned methods, some efforts [29, 27] employ neural representations of 3D human mesh points in a canonical pose to model human appearance details, thereby achieving multi-view and temporal consistency in human videos. However, these methods primarily focus on appearance consistency and full-body pose accuracy, overlooking the importance of facial details and hand movements for video expressiveness and specific scenario suitability.

## 3  Method

The overview of our proposed framework ShowMaker is shown in Fig. 1, which can achieve high-quality video-driven conversational avatar synthesis. In the following, we first define our task in Sec. 3.1 and provide an overview of our framework pipeline in Sec. 3.2. Then detailed explanations of our novelly designed modules are demonstrated in Sec. 3.3 and Sec. 3.4. Finally, the training strategies are introduced in Sec. 3.5.

### 3.1  Task Formulation

For a conversational video $V$, we leverage the pre-trained DWPose [50] to extract human body key points of all frames, including face, body, and hands, and then we paint them into 2D heatmaps $P$ as driving pose. Given a reference image $I_r \in \mathbb{R}^{H \times W \times 3}$ from a target person, and the driving poses $P_d = \{P_1, P_2, \ldots, P_F\} \in \mathbb{R}^{F \times H \times W \times 3}$, our task aims at generating target video sequence $\hat{V} = \{\hat{I}_1, \hat{I}_2, \ldots, \hat{I}_F\} \in \mathbb{R}^{F \times H \times W \times 3}$ with similar facial and body movements as in the driving poses $P_d$. The entire generative process can be formulated as $\hat{V} = \mathcal{G}(I_r, P_d)$, where $\mathcal{G}$ represents the generative model.

### 3.2  Pipeline Overview.

Fig. 1 provides an overview of our approach. We follow the recent works [1, 13, 47, 56] to adopt a dual-stream learning paradigm. The upstream branch (Green Area) processes the reference image, extracting the appearance information using a VAE encoder and Reference U-Net. In the downstream branch (Blue Area), the Pose Encoder takes the 2D pose heatmaps as input to extract human pose information, including face and hand. The subsequent Denoising U-Net effectively integrates the extracted appearance and driving information to predict noise intensity. Additionally, we introduce a novel Key Points-based Fine-grained Hand Modeling module to capture robust hand structure and texture features, and the comprehensive face representations are constructed by the Face Recapture module. The Denoising U-Net is derived from the pre-trained SD [33], replacing self-attention with reference-attention and CLIP [31] cross-attention with face attention, where hand attention integrates hand features from the Key Points-based Fine-grained Hand Modeling module, and face attention incorporates face features from the Face Recapture module. Additionally, temporal attention layers are introduced to promote the temporal consistency of the generated images.

**Reference U-Net.** Numerous studies have demonstrated that the U-Net network outperforms the CLIP image encoder in terms of appearance extraction. Firstly, the image resolution input to CLIP is limited to $224 \times 224$, which is insufficient for capturing detailed appearance information. Secondly, the features extracted by the CLIP image encoder are primarily oriented towards high-level semantic matching. We thus follow recent works [1, 13, 47, 56] to adopt a duplicate U-Net as our appearance extraction network named Reference U-Net, initializing its weights from the pre-trained SD.

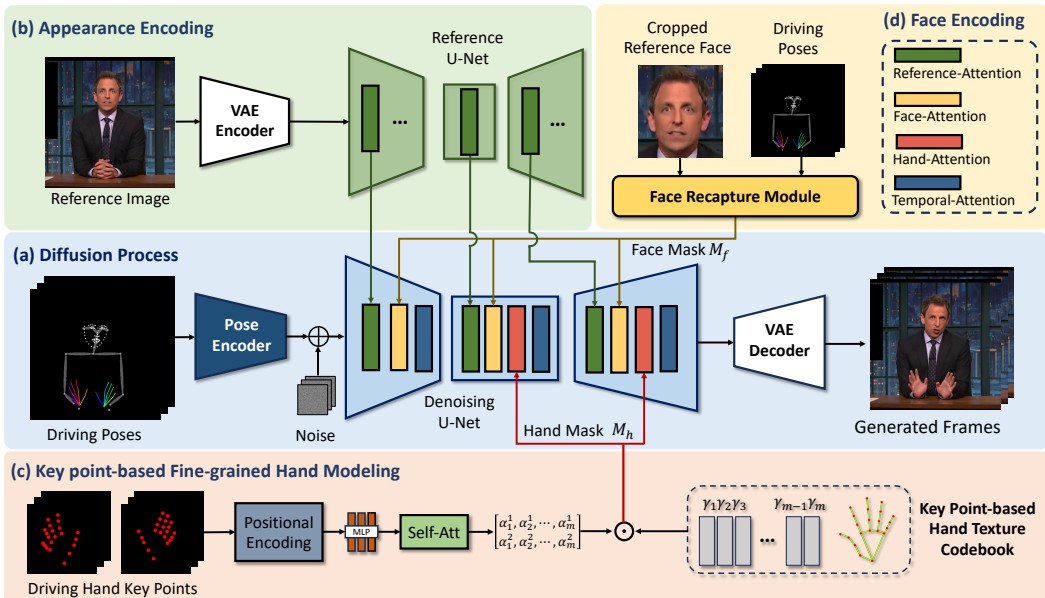

Figure 1: **Overview of our proposed framework ShowMaker.** Our framework adopts a dual-stream design including a Reference U-Net and a Denoising U-Net, where the former takes a reference image as input for appearance encoding and the latter takes noise latent and driving poses as input for diffusion processing. We further equip the backbone with a Key Point-based Fine-grained Hand Modeling module and a Face Recapture module for fine-grained avatar synthesis.

Specifically, as shown in Fig. 1, the reference image is first encoded into the latent space via the VAE Encoder and then passed to the Reference U-Net. The feature map $\mathbf{z}_r$ from each layer in Reference U-Net is utilized in the reference attention mechanism for appearance detail fusion. Initially, $\mathbf{z}_r$ is repeated $F$ times along the temporal dimension to form $\mathbf{z}_r \in \mathbb{R}^{B \times F \times h \times w \times c}$ to match the size of the denoising feature map. Here, $B$ indicates the batch size.

**Pose Encoder.** To reduce computational complexity, we employ a lightweight network as the pose encoder which consists of 8 convolutional layers initialized with Gaussian weights, with the final layer utilizing zero convolution. The pose information extracted by the pose encoder is then added to the noise latent and fed into the Denoising U-Net.

### 3.3 Key point-based Fine-grained Hand Modeling

Although the driving pose provides important guidance for hand synthesis, the hand regions only occupy a limited number of pixels. The convolution operation in the pose encoder and Denoising U-Net repeatedly downsamples the spatial size, which progressively weakens this guidance and results in unexpected structural and textural artifacts. Considering hand key points are clearly defined and cannot be constrained by the resolution of the hand region, we make our attempt to enhance this guidance by involving absolute coordinates of hand key points as additional inputs. Specifically, given the $m$ key points of one hand $K = [k_1; k_2; \ldots; k_m]$, we first use Fourier positional encoding [26] to map their coordinates into a high-dimensional space, which can capture subtle differences in hand gestures. Formally, the frequency function $\mathcal{F}$ in positional encoding is defined as:

$$\mathcal{F}(k) = [\left(\sin\left(2^0\pi k\right), \cos\left(2^0\pi k\right), \cdots, \sin\left(2^{L-1}\pi k\right), \cos\left(2^{L-1}\pi k\right)\right), k]. \tag{1}$$

Here $k \in \mathbb{R}^2$ is the coordinate value of a single hand key point, which has been normalized to $[0, 1]$. Notably, we retain the original coordinate value in Eq. 1 and we set $L = 20$ by default. By using the function $\mathcal{F}$, the coordinates of hand key points $K_h \in \mathbb{R}^{B \times F \times 2 \times 2m}$ can be mapped to high dimensional space $\mathbf{F}_h \in \mathbb{R}^{B \times F \times 2 \times 2m*(2L+1)}$, which enables reliable enhancement on the guidance from the raw driving poses $P_d$. Here, the first 2 denotes two hands (both left and right hand).

Based on the enhanced hand structure guidance, we additionally design a key point-based hand texture codebook that achieves fine-grained hand texture synthesis. Particularly, as illustrated in Fig. 1 (c), the key point-based hand texture codebook consists of a set of learnable basis vectors $\mathbf{C}_{hand} = \{\gamma_i\}_{i=1}^m, \gamma_i \in \mathbb{R}^d$, which is built in the same topology as the $m$ hand key points mentioned above. It is worth noting that though the left hand and right hand are symmetric in geometry, they share the same topology (i.e., the order of key points is the same). Moreover, these basis vectors are constrained to be orthogonal so that each basis vector represents a distinct hand texture pattern and only requires an emphasis on the texture modeling around its corresponding key point. Specifically, any two basis vectors $\gamma_i, \gamma_j$ satisfy the following conditions:

$$< \gamma_i, \gamma_j > = \begin{cases} 0 & i \neq j, \\ 1 & i = j. \end{cases} \tag{2}$$

Subsequently, a self-attention layer followed by a linear projection layer is employed to predict the weights of hand texture patterns $\mathbf{A} \in \mathbb{R}^{B \times F \times 2 \times m}$ from $\mathbf{F}_h$. By applying a weighted combination on all the basis vectors, we obtain the key point-based hand texture compensation defined as:

$$\mathbf{G}_h = \sum_{i=1}^m a_i \gamma_i. \tag{3}$$

Finally, the output feature $\mathbf{G}_h \in \mathbb{R}^{B \times F \times 2 \times d}$ is sent to the hand attention layers in the Denoising U-Net for cross-attention calculation. Additionally, to explicitly inject hand motion information into the denoising latent feature maps, we apply a hand mask $M_h$ to provide emphasis guidance on the hand regions, which can be expressed as:

$$latents = \text{Att}_{hand}(latents, \mathbf{G}_h, \mathbf{G}_h) * M_h + latents,$$
$$\text{Att}_{hand}(latents, \mathbf{G}_h, \mathbf{G}_h) = \text{Softmax}\left(\frac{(\mathbf{W}_Q \cdot latents)(\mathbf{W}_K \cdot \mathbf{G}_h)^\top}{\sqrt{d}}\right) \cdot (\mathbf{W}_V \cdot \mathbf{G}_h). \tag{4}$$

where $latents$ denotes the denoising latent feature maps, $\mathbf{W}_Q, \mathbf{W}_K, \mathbf{W}_V$ represent learnable weights for the cross-attention modules.

### 3.4 Face Recapture Module

Similar to the challenge with hand generation, producing a satisfactory face in human video synthesis is difficult. Most approaches rely on post-processing strategies to address this issue, but these methods significantly increase training and inference costs. In this paper, we design a Face Recapture module to extract comprehensive face information and inject it into the face attention layer in the Denoising U-Net to improve the quality of the generated faces.

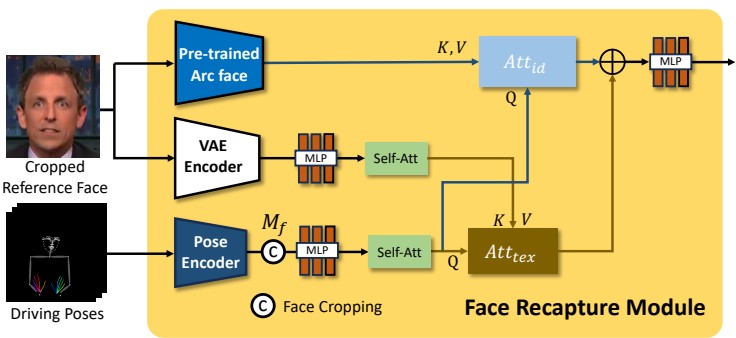

Figure 2: **Architecture of the proposed Face Recapture Module.**

Specifically, as shown in Fig. 2, we enhance the face area of the generated image from two aspects: texture details and global identity. First, we use a face detection model to crop and align the face from the reference image. Next, the pre-trained VAE Encoder and face recognition model ArcFace are leveraged to extract the facial texture feature $\mathbf{F}_{tex}$ and global identity feature $\mathbf{F}_{id}$, respectively. For the facial texture feature, we perform feature enhancement by combining a MLP and a self-attention

layer. We then repeat these features $F$ times along the temporal dimension to match the size of driving poses and produce $\mathbf{F}_{id} \in \mathbb{R}^{B \times F \times 1 \times d_{id}}$ and $\mathbf{F}_{tex} \in \mathbb{R}^{B \times F \times (hw) \times d_f}$.

In the meantime, we crop out the corresponding facial pose information $\mathbf{F}_d$ from the driving pose feature maps according to the face mask $M_f$. Similarly, a MLP and a self-attention are proposed for feature enhancement on $\mathbf{F}_d \in \mathbb{R}^{B \times F \times (hw) \times d_f}$. In order to equip $\mathbf{F}_d$ with textural and identity information, we adopt two separate cross-attention blocks to embed texture and identity information into the $\mathbf{F}_d$, respectively. Specifically, by taking $\mathbf{F}_d$ as query, $\mathbf{F}_{tex}$ as key and value, we define the texture fusing process as $\mathbf{G}_{tex} = \text{Att}_{tex}(\mathbf{F}_d, \mathbf{F}_{tex}, \mathbf{F}_{tex})$. Similarly, by taking $\mathbf{F}_d$ as query, $\mathbf{F}_{id}$ as key and value, we can obtain the fused identity feature $\mathbf{G}_{id}$ through $\mathbf{G}_{id} = \text{Att}_{id}(\mathbf{F}_d, \mathbf{F}_{id}, \mathbf{F}_{id})$. The comprehensive face information is obtained by

$$\mathbf{G}_f = \text{MLP}(\mathbf{G}_{tex} + \mathbf{G}_{id}). \tag{5}$$

Finally, $\mathbf{G}_f \in \mathbb{R}^{B \times F \times (hw) \times d_f}$ is injected into the face attention layer in the Denoising U-Net. Similarly, the face mask is utilized to constrain the latent feature maps after face attention:

$$latents = \text{Att}_{face}(latents, \mathbf{G}_f, \mathbf{G}_f) * M_f + latents. \tag{6}$$

### 3.5 Training strategies

During the diffusion process, the random noise is continuously added to the real image until it reaches a state of Gaussian noise. Conversely, the generation process within the diffusion model is the inverse of the diffusion process, which takes random Gaussian noise as input and generates real images through gradual denoising. In the training process, the diffusion model leverages the Denoising U-Net to predict the added noise at various time steps and the optimization objective can be defined as:

$$L = \mathbb{E}_{\mathbf{z}_t, \mathbf{c}, \epsilon, t} \left( \| \epsilon - \epsilon_\theta \left( \mathbf{z}_t, \mathbf{c}, t \right) \|_2^2 \right), \tag{7}$$

where $\mathbf{c}$ is the conditional features, $\epsilon_\theta$ represents the Denoising U-Net, $\mathbf{z}_t$ is the denoising latent feature maps at timestep $t$.

In our work, we adopt a two-stage training strategy to separately perform appearance modeling and temporal consistency modeling. The first stage is image-level training. During this stage, we set $F = 1$, the training goal is to accurately map the appearance details from the reference image to the driving pose. The VAE encoder and CLIP Image encoder are fixed, and the rest of the network except for the temporal attention are all trainable. In terms of the second stage, the objective is to improve the temporal coherence of the generated frames. During this stage, we only perform training on the temporal attention layer with all other network weights fixed and the weights for the temporal attention layers in the Denoising U-Net are initialized by the pre-trained AnimateDiff [8]. The loss function can be reformulated as:

$$L = \mathbb{E} \left( \| \epsilon - \epsilon_\theta \left( \mathbf{z}_t, \mathbf{z}_r, \mathbf{G}_h, M_h, \mathbf{G}_f, M_f, t \right) \|_2^2 \right), \tag{8}$$

where $\mathbf{z}_r$ represents the appearance feature maps extracted by Reference U-Net, $\mathbf{G}_h$ represents the hand movement feature maps from the Key Point-based Fine-grained Hand Modeling module, and $\mathbf{G}_f$ is the face feature maps obtained by the Face Recapture Module. Additionally, $M_h$ and $M_f$ denote the masks for the face and hand regions, respectively.

It is worth noting that in order to focus on the face and hand area generation, in the later phase of the first training stage, we adopt the hand mask and face mask to calculate the $L_1$ loss of the corresponding area as the final loss every 10 iterations.

## 4 Experiments

### 4.1 Experimental Settings

**Datasets.** In order to verify the effectiveness of the proposed method, we select the videos of two actors, Seth and Oliver, in the talkshow [51] dataset for training and testing. In addition, to enrich the diversity of characters and hand movements, we record videos of seven people in the indoor scene. The video length of each person is about 10 minutes and we divide the videos into multiple clips of

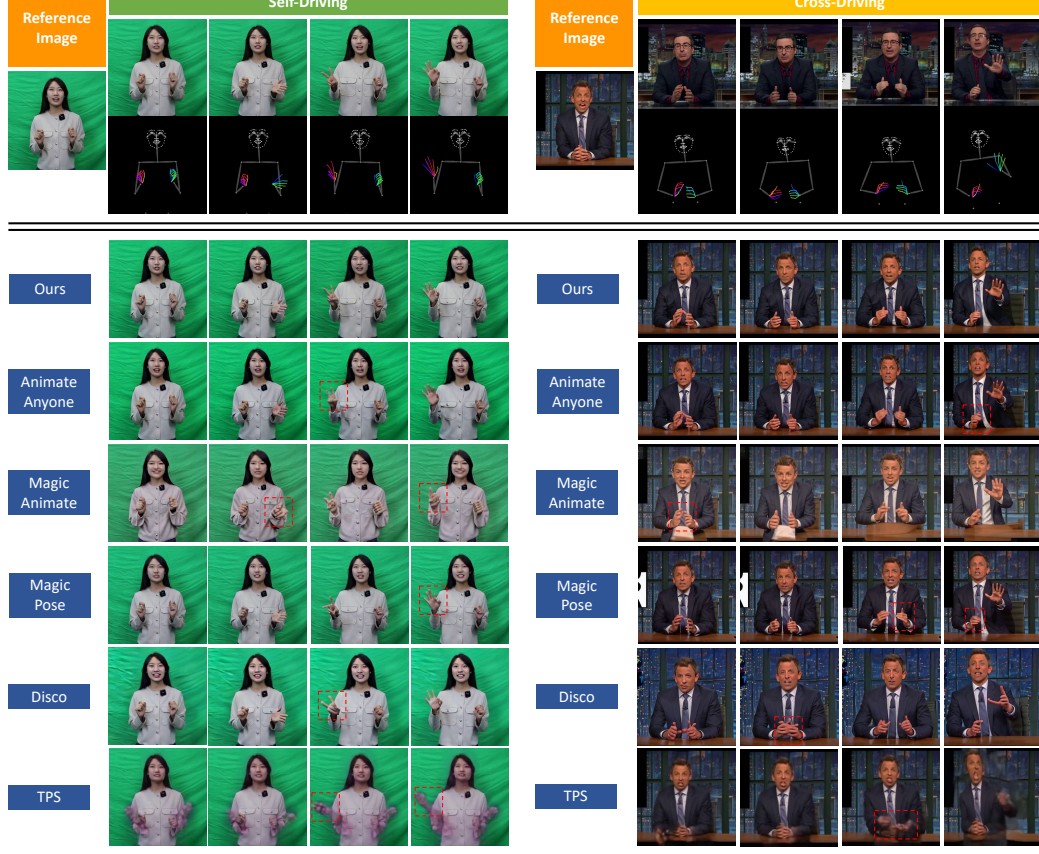

Figure 3: Qualitative results compared with other methods. Our approach achieves high-fidelity gesture details and satisfactory image quality in both self and cross-driving settings.

about 8 seconds for training convenience. In addition to simple rhythmic gestures, these recorded videos also include many complex gestures such as numbers. We randomly divide the training set and test set according to 9:1.

**Implementation Details.** For data processing, we crop out the body region with a resolution of $512 \times 512$ and utilize the pre-trained face detection model [2] to crop and align faces following FFHQ [20]. The resolution of the face image is $256 \times 256$. The hand and face masks are determined based on the largest circumscribed rectangle of the corresponding key points. All experiments were completed on 8 A800s, with a learning rate of 1e-5. For the first training stage, the batch size $B$ is set to 24, and sequence length $F$ is set to 1. The training step is 100k, which takes about six days. For the second stage, $B$ and $F$ are set to 1 and 24, respectively with a 30k training step, taking about one day. During inference, we adopt a CFG [10] of 7.5 and perform 30 denoising steps using the DDIM sampler. Additionally, to ensure temporal consistency between different clips of the same video, we use the same reference image, with an overlap of 4 frames between adjacent driving clips.

**Comparison Methods.** We select several state-of-the-art approaches, including TPS [54], Disco [44], AnimateAnyone [13], MagicAnimate [47], MagicPose [1], Make-Your-Anchor [17] as our comparison methods. TPS is a general animation model based on warping operation, which adopts GAN as the backbone, the remains are diffusion-based human video generation models. Among them, Make-Your-Anchor provides the specific person generation model. We finetune these models on the same training dataset with the official codes and pre-trained models.

**Metrics.** We comprehensively measure the quality of generated images from pixel space and feature space using SSIM, PSNR, and FID [9] and adopt FVD [38] to verify the temporal consistency of generated videos. In addition, the motion accuracies of body ($L_{body}$), face ($L_{face}$), and hand ($L_{hand}$) are measured by calculating the mean Euclidean distance between the key points of the generated images and real images.

Table 1: Quantitative results of our approach compared with SOTAs. Our method achieves the best performance on image quality, temporal consistency, and motion precision.

| Method | SSIM ↑ | PSNR ↑ | FID ↓ | FVD ↓ | $L_{body}$ ↓ | $L_{face}$ ↓ | $L_{hand}$ ↓ |
|---|---|---|---|---|---|---|---|
| TPS | 0.65 | 29.02 | 94.77 | 1120.37 | 5.99 | 1.26 | 17.99 |
| Disco | 0.69 | 29.13 | 80.76 | 540.76 | 5.85 | 1.52 | 4.33 |
| AnimateAnyone | 0.80 | 29.41 | 16.87 | 365.83 | 2.73 | 0.62 | 1.10 |
| MagicAnimate | 0.70 | 28.55 | 50.24 | 665.21 | 4.48 | 1.33 | 3.02 |
| MagicPose | 0.82 | 30.03 | 16.37 | 370.75 | 2.32 | 0.68 | 1.12 |
| Ours | **0.85** | **32.23** | **15.43** | **197.43** | **2.27** | **0.19** | **0.77** |
| Make-Your-Anchor (Seth) | 0.63 | 29.18 | 32.32 | 428.84 | 4.55 | 1.07 | 1.64 |
| Ours (Seth) | **0.85** | **33.14** | **9.83** | **193.25** | **2.10** | **0.21** | **0.72** |

## 4.2 Comparison with Other Methods

**Quantitative Results.** The quantitative results of our methods compared with SOTAs are shown in Tab. 1. Our method achieves the best results on SSIM, PSNR, and FID metrics, underscoring its clear advantages on image quality. In addition, our method obtains the best FVD indicating that the temporal consistency of generated videos is superior. Moreover, our approach outperforms others on the motion accuracy metric including ($L_{body}$), face ($L_{face}$), and hand ($L_{hand}$) which demonstrates the precision of generated poses and gestures. Benefiting from the proposed Key Points-based Fine-grained Hand Modeling module and Face Recapture module, our framework is able to generate accurate hand gestures and high-fidelity facial details. TPS is a general object animation approach and struggles with complex human poses and hand postures, resulting in the poorest overall generation quality. Disco, AnimateAnyone, MagicAnimate and MagicPose focus on generating coarse-grained human poses without fine-grained modeling of hands and faces, leading to lower generation quality, particularly in hand gesture accuracy. Note that Make-Your-Anchor [17] only provides a pre-trained model on Seth data and thus we also report the experimental results on the Seth for a fair comparison. It is observed that our method outperforms [17] on all metrics.

**Qualitative Comparison.** For qualitative comparison, we conduct two experimental settings, including self-identity and cross-identity driven. The self-identity driven comparison results are shown on the left of Fig. 3 and 4, where obvious artifacts are marked with red dotted boxes. Our method is able to generate accurate gestures and high-quality hand details while other approaches fail to convey complex gestures. In addition, face identity is well-preserved in our results, while other methods lead to unnatural face shapes, textures, and identity. Note that cross-identity driven is more challenging and our method still achieves high-fidelity gesture details and satisfactory image quality as indicated in the right of Fig. 3 and 4. The hand areas of other results are blurry and unrealistic. In summary, our method makes use of hand and face information through the well-designed Key Points-based Fine-grained Hand Modeling module and Face Recapture module, enhancing the video quality which could meet complicated requirements in conversational scenarios.

**Human Evaluation.** We conduct a user preference study to evaluate the performance of the proposed framework. There are 21 samples and 15 human voters in total. For each sample, we present six video results generated with ShowMaker and other SOTA methods to the human voter in a random order. The human voters are required to estimate the video results in three aspects: a) Motion Accuracy: Does the video accurately reproduce the motion in the driving video? b) Appearance Consistency: Does the subject in the video have a consistent appearance with the reference image? c) Temporal Consistency: How is the temporal coherence of this video? The rating score ranges from 1 to 5 and higher scores indicate better preference. As shown in Tab. 3, our method achieves the highest scores compared with its counterparts, which demonstrates that our method is preferred by a significant margin on motion accuracy and appearance consistency.

## 4.3 Ablation Study

To verify the contributions of different components, we train three variants by removing the Key Points-based Fine-grained Hand Modeling module (HM), the Face Recapture module (FR), and two-stage training (Stage 2), respectively.

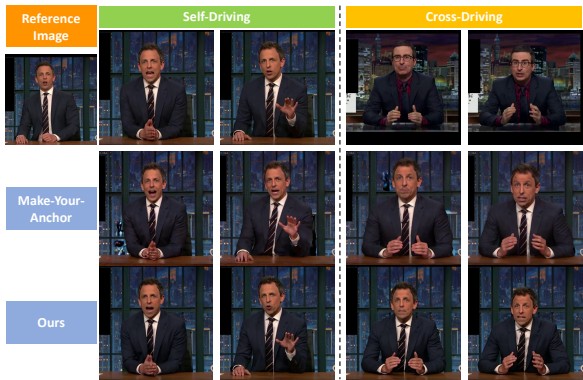

Figure 4: Qualitative comparison between Make-Your-Anchor and our ShowMaker.

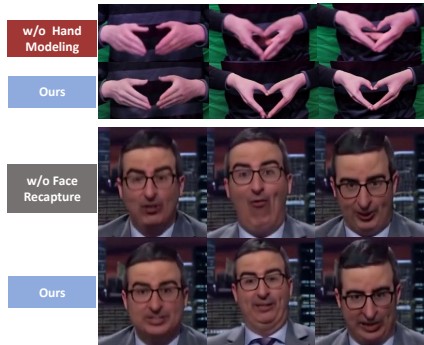

Figure 5: Qualitative ablation results when removing different components in our framework.

Table 2: Ablation analysis of HM, FR, and two-stage training.

| Variations | FVD ↓ | $L_{face}$ ↓ | $L_{hand}$ ↓ |
|---|---|---|---|
| w/o HM | 212.25 | - | 1.12 |
| w/o FR | 208.28 | 0.23 | - |
| w/o Stage 2 | 373.87 | - | - |
| Ours | **197.43** | **0.19** | **0.77** |

Table 3: User Study.

| Methods | Ours | Magic Pose | Magic Animate | Animate Anyone | Make-Your-Anchor | Disco | TSP |
|---|---|---|---|---|---|---|---|
| Motion Accuracy | **4.53** | 3.13 | 2.87 | 3.69 | 4.43 | 2.62 | 1.84 |
| Appearance Consistency | **4.23** | 3.47 | 2.00 | 3.62 | 4.18 | 2.16 | 1.91 |
| Temporal Consistency | **4.33** | 3.00 | 2.20 | 3.11 | 3.57 | 1.78 | 1.29 |

For quantitative results, we report the FVD and motion accuracy metrics in Tab. 2. It is observed that the HM and FR could enhance gesture accuracy and face quality by a significant margin. The two-stage training brings about notable improvements in FVD which indicates better temporal consistency. Additionally, we present qualitative comparisons to verify the effectiveness of the proposed HM and FR module. As shown in Fig. 5, the generated hands are more satisfactory, and the face shapes and textures are realistic and clear with the proposed modules.

## 5    Conclusion and Discussion

**Conclusion.** In this paper, we propose the framework ShowMaker, which achieves high-fidelity 2D human video synthesis with two novel designs to achieve fine-grained diffusion modeling. We first propose a Key Point-based Fine-grained Hand Modeling module for robust and fine-grained hand synthesis which takes advantage of 2D hand key points and a key point-based codebook. To further reconstruct the facial details and identity information, we introduce a Face Recapture module, which effectively equips the structure information with detailed textural information and global identity. Quantitative and qualitative evaluation has indicated the superiority of our framework beyond the existing approaches.

**Limitations.** Despite the success of our framework, we also recognize some limitations during the exploration. Firstly, our Key Point-based Fine-grained Hand Modeling module can robustly manage hand synthesis with occasional incorrect hand gestures. However, DWPose [50] suffers from performance degradation when handling videos with severe motion blur leading to considerable perturbation in the driving signal, which inevitably results in unexpected artifacts in our results. Secondly, our framework produces less satisfactory results when handling challenging cases, such as reflection on glasses. These will be part of our future work.

**Broader Impact.** Our method focuses on synthesizing realistic 2D avatars with rich facial expressions and complex hand gestures, which is intended for developing digital humans under more daily scenarios like TV shows. However, it may also be misused for some malicious purposes on social media, which leads to negative impacts on the whole society. Therefore, we will make our efforts to strictly oversee the dissemination of our models as well as the resulting content and also restrict access solely to research-oriented demands. We believe that the proper use of this technique will enhance positive societal development in both machine learning research and daily life.

## ACKNOWLEDGMENTS

This work is supported by the National Nature Science Foundation of China (62121002, U23B2028, 62232006, 62032006, 62472395).

We acknowledge the support of the GPU cluster built by the MCC Lab of Information Science and Technology Institution, USTC. We also thank the USTC Supercomputing Center for providing computational resources for this project.

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

# A Appendix

## A.1 Pose Encoder Structure

Fig. 6 shows the detailed network structure of the pose encoder.

## A.2 Dataset Details

Tab. 4 gives the number of training and testing clips for the indoor recording dataset and talkshow dataset. The length of each clip is 2-15 seconds.

When preparing the training data, we first produce the DWPose results from each frame and set the center of shoulders as the cropping center. Then we crop the original video frame using an adaptive cropping size, where the cropping size is designed as a fixed ratio (we set it to 2.65) of the shoulder width. This operation forces the human body to lie in a roughly consistent position. Finally, all cropped images are resized to $512 \times 512$.

In the inference stage, for the cross-identity driven setting, we further bridge the gap between body shapes by scaling the driving poses to match the reference pose. The scaling ratios of width and height are defined as $W_r/W_d$, and $H_r/H_d$, where $W$ and $H$ represent the shoulder width and the height of the human body, respectively. The shoulder width is measured as the Euclidean distance between the left and right shoulder key points, while the body height is determined by the Euclidean distance from the nose key point to the pelvis key point.

## A.3 More Ablation Experiments

To further verify the effectiveness of the proposed Key point-based Fine-grained Hand Modeling (HM), we show more ablation experiments in Fig. 7, where **Vanilla** refers to removing the entire HM module from our proposed ShowMaker, **Hand image** refers to adopting the cropped hand image to extract texture features through the VAE encoder and then feeding it into the hand attention, and **w/o positional encoding** refers to not using positional encoding in the HM module. It can be seen that using the hand image as compensation information has no obvious effect on improving the hand texture and structure, As for HM, positional encoding can significantly improve the high-frequency details and structural accuracy of the generated hands.

## A.4 Temporal consistency

Fig. 8 shows two generated video sequences, each sequence contains 3 adjacent video frames. It can be seen that the generated video frames at different times have consistent appearance and no temporal texture changes occur.

## A.5 Challenging Examples

Fig. 9 shows two challenging sample generations. In the example on the left, the driving pose is extracted from a female video, while the reference image is a male, and the driving gestures and the reference gesture are very different. In the example on the right, the reference image is not a common frontal image instead of a side image. Our proposed ShowMaker can still generate target frames well in these two cases, which shows that our method has satisfied robustness.

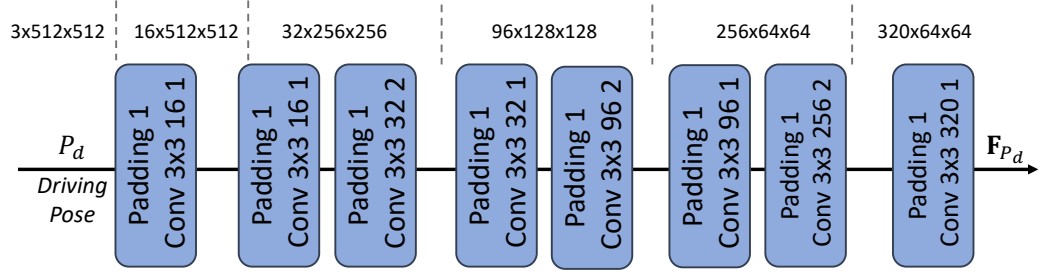

Figure 6: The pose encoder structure consists of 8 convolutional layers, where (Conv $3 \times 3$ 16 1) represents the kernel size is $3 \times 3 \times 16$, and the stride is 1. Except for the last convolutional layer, each convolution is followed by GroupNormalization and the activation function silu.

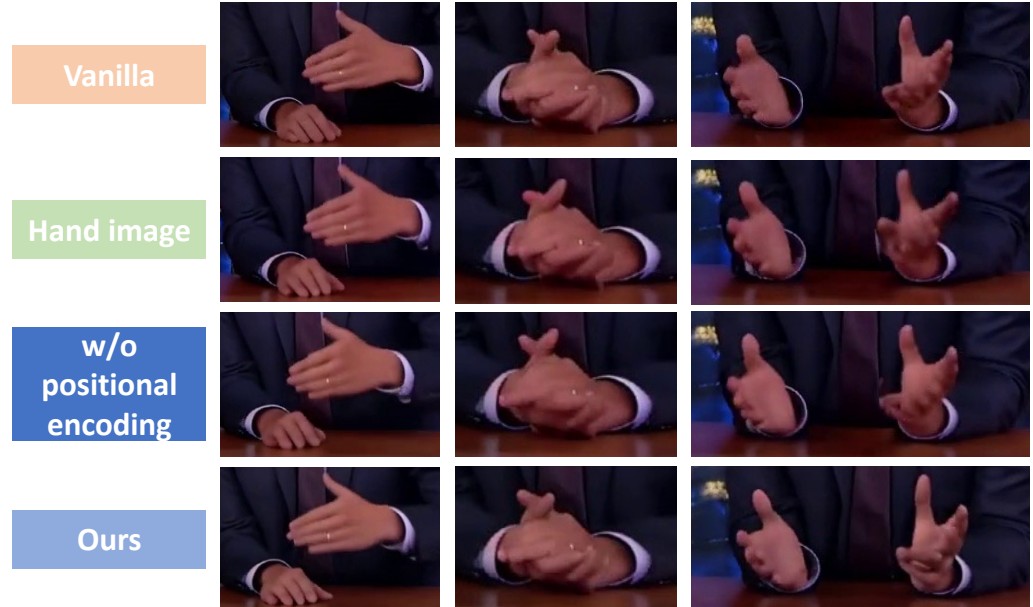

Figure 7: Ablation experiments of hand module.

Table 4: Dataset details. ID1-7 are datasets recorded indoors. Seth and Oliver belong to the talkshow dataset. All training and test sets do not overlap.

| IDs | The number of clips in the training set | The number of clips in the test set |
|---|---|---|
| ID1 | 49 | 5 |
| ID2 | 57 | 6 |
| ID3 | 56 | 6 |
| ID4 | 65 | 7 |
| ID5 | 47 | 5 |
| ID6 | 64 | 7 |
| ID7 | 55 | 6 |
| Seth | 3306 | 100 |
| Oliver | 6746 | 200 |

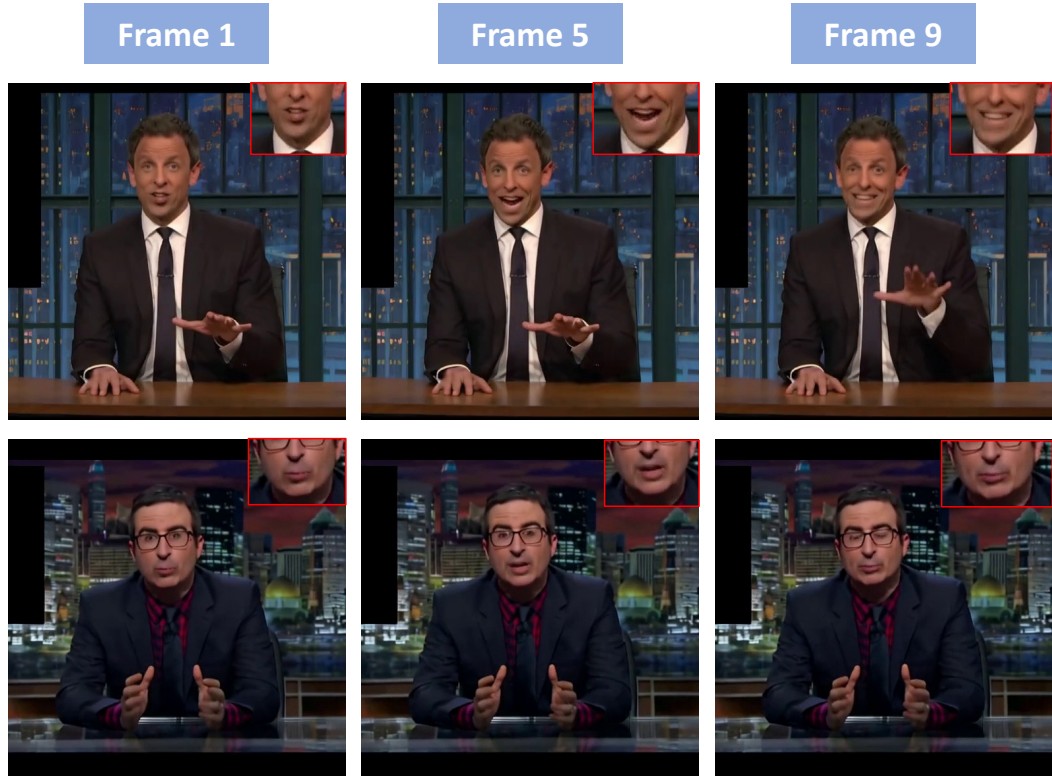

Figure 8: The generation results of adjacent frames.

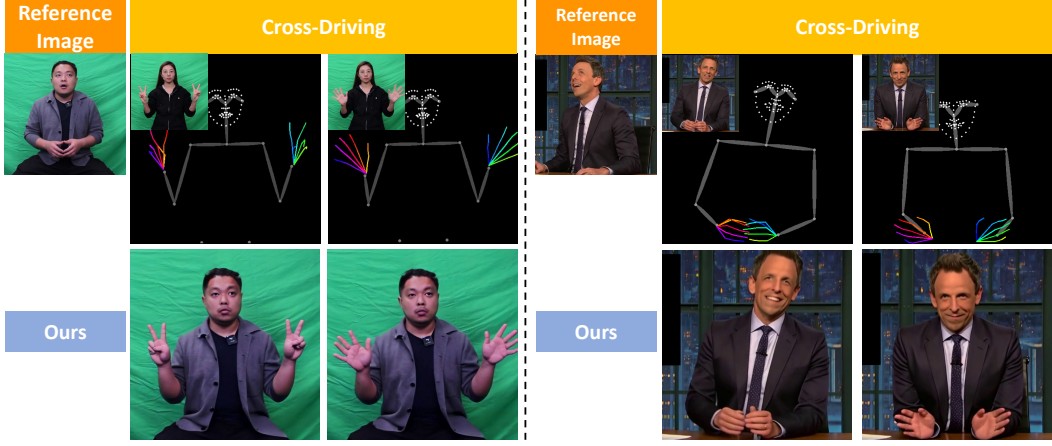

Figure 9: The generated results when the pose difference is obvious.

