# OpenReview forum: "ShowMaker: Creating High-Fidelity 2D Human Video via Fine-Grained Diffusion Modeling"
_NeurIPS.cc/2024/Conference — NeurIPS 2024 poster_

### Official Review · Reviewer_xzB4 · 2024-06-17

**Soundness:** 3
**Presentation:** 3
**Contribution:** 2
**Rating:** 6
**Confidence:** 5

**Summary:**

This paper proposed a pipeline for 2D human motion retargeting, based on 2D key points and face encoding via fine-grained diffusion modeling.

**Strengths:**

The paper is well-written, and the demo presents satisfying hand and face modeling results.

**Weaknesses:**

1. Missing recent baselines in the experimental comparison. E.g. Diffusion-based method: MagicPose from ICML 2024, GAN-based methods: TPS from CVPR2022 etc.

2. The novelty of the proposed method is limited. It seems to me that the only novel component of this paper compared to previous works is the face encoding and key point-based hand modeling. The ReferenceNet for identity control and Pose Encoder for pose control are all from previous works (ReferenceNet from Animate Anyone, MagicAnimate, MagicPose, Champ, and ControlNet from DisCo, MagicAnimate, Animate Anyone, MagicPose). Temporal Attention the same.

3. I would like to see the performance of the proposed method compared to baselines on the TikTok dataset since this is a public dataset, and all previous works reported metrics on it. I'm not implying the authors are hiding something but I think it's necessary to follow previous settings and report the performance on the new dataset (talkshow).

4. Lack of Long Video Generation Results. The presented videos in the supplementary materials are very short compared to MagicAnimate's videos on TikTok dataset. It's necessary to provide longer visualization.

5. Lack of motion retargeting visualization while the condition of human DW-Pose is significantly different than the Human Pose in the reference image. The visualization in both supp and main paper shows only the cases where condition and reference image share similar human poses. This also raises the concern about face encoding and key point-based hand modeling: will such network design preserves human identity and ensure temporal coherence when the poses have a huge gap between reference and condition?

**Questions:**

See weakness. Major questions as following:

1. Examples of Long Video Generation. Ideally, the generated video of the TikTok test set as MagicAnimate shows. (~at least 20 seconds or longer)

2. Motion retargeting visualization while condition and reference have quite different human poses. Please see the previous paper, MagicPose Figure 5,7,17, for reference.

3. Previous work, e.g., MagicPose, has shown that the model can synthesize the back of the human subject. Can this model do the same? Given that the front of the human is the reference image and another pose condition, how can the model learn to distinguish the front/back of the human from such conditions?

4. Can this model provide a satisfying result if the identity reference is a non-frontal image,e.g., the left or right side of the human?

5. Is there any visualization on out-of-domain motion retargeting? E.f. cartoon-style animation?

6. More detailed explanation of the novelty.

**Limitations:**

The authors include a limitation section.

---

> ### Author Rebuttal · Authors · 2024-08-07
>
> # Response to Reviewer xzB4
> Thanks for such a detailed review. Here are my responses.
>
> According to the provided reviews, we believe there are a few misunderstandings on the task setting that need to be clarified.
>
> __Task \& Setting Issues__
>
> &emsp; (1) Unlike the approaches concentrating on holistic body movement synthesis (e.g., AnimateAnyone and MagicPose), our ShowMaker focuses on __half-body 2D talking avatar__ creation,  which aims to upgrade the traditional 2D talking head avatar task to the next level with richer movements (e.g., hand gestures and upper body movements). In this scenario, the training data usually contains frontal images with very few side-view cases, therefore learning to distinguish the front/back of the human is not necessary (Question 3).
>
> &emsp; (2) Our approach follows the __person-specific setting__ as Make-Your-Anchor rather than the one-shot animation setting used in AnimateAnyone. We provide our train/test data information in Table 1 in the pdf file. Under the person-specific setting, we focus on self-driving and cross-driving performance on trained IDs instead of out-of-domain data (e.g., cartoon-style images (Question 5)). Considering this person-specific setting and our task scenario, we conduct experiments on the TALKSHOW dataset and our collected dataset instead of the TikTok dataset (Weakness 3).
>
> &emsp; (3) The dual-branch design with a ReferenceNet is an appropriate backbone for generative tasks, but we did not involve them in our contributions. Our technical contributions are summarized below (Weakness 2 and Question 6):
>
> &emsp; &emsp; (a) We propose an effective hand recovery module for fine-grained synthesis by using a positional encoding strategy and a carefully designed key point-based texture codebook. Our approach shows obvious superiority in the hand synthesis over other dual-branch baselines.
>
> &emsp; &emsp; (b) We propose a Face Recapture module to improve facial details and identity-preserving ability, especially under the cross-driving setting with pose and shape gaps. The results in our supplementary video and the pdf file also show the temporally consistent facial texture and identity.
>
> In addition to the task and setting issues, there are also some experimental issues.
>
> __Experimental Issues__
>
> - Q1. Comparison with TPS from CVPR2022 and MagicPose from ICML 2024. (Response to  Weakness 1)
>
>   A1. Thanks for your suggestion. We compared the TPS method in our paper, please see Table 1 and Figure 3 in the main paper for more details. In terms of MagicPose, since it is a concurrent work that was accepted by ICML in May 2024 (very close to the submission date), we involve another work MagicAnimate (similar to MagicPose) in comparison.
>
> &emsp; Here, we finetune MagicPose on our training data and provide the comparison results in Figure 4 in the pdf file. Similar to AnimateAnyone and MagicAnimate, MagicPose suffers from inaccurate generation and texture degradation in both facial and hand regions. Please zoom in and see the details in the red boxes. The results demonstrate the necessity of our proposed Face Recapture module and Fine-Grained Hand Modeling.
>
> - Q2. About longer visualization. (Response to Question 1)
>
>   A2. In our supplementary video, we have provided demo videos of 10-15 seconds, which is comparable to the length of videos in the TikTok dataset.  During our exploration, we managed to generate videos longer than 30 seconds with high-fidelity and temporally consistent texture. Since we cannot provide external links, we will add longer-generated videos to our supplementary video.
>
> - Q3. Pose gaps between the reference and driving images & Non-frontal images. (Response to Question 2, Question 4)
>
>   A3. We provide two examples in Figure 5 in the pdf file to illustrate how our ShowMaker performs when dealing with challenging cases with pose gaps or non-frontal images.
>
>   &emsp; (1) For pose gaps, we first provide an example by using a reference image from a male and driving signals from a female. As shown in Figure 5 (left part), there is a huge gap in both pose and shape. Please also refer to A4 for Reviewer jdWr, we adopt our pose alignment strategy to first align the driving signals with that of the reference image for better synthesis and the results show the effectiveness of our approach under huge pose gaps.
>
>   &emsp; (2) Non-frontal images. Since most of the videos in our dataset are in the front view, so we took some time to look for a side-view reference image in the test set, and the generated result is shown in Figure 5 (right part). The results demonstrate that our network design can preserve human identity and hand texture even when processing non-frontal reference images.

---

> > ### Comment · Reviewer_xzB4 · 2024-08-13
> >
> > Thanks for the author's detailed response to my questions. My concern on visualization and experiments has been well-addressed. Hence, I'm raising my score.
> >
> > For out-of-domain data, I still suggest the authors explore training/finetuning the trained model on real-human data on **half-body 2D talking avatar** of cartoon-style videos since it would be an outstanding contribution to demonstrate the generalization ability, and it would be interesting to see the results.
> >
> > For MagicPose ICML 2024, kindly remind that it's released around Sep 2023, which is close to Animate-Anyone/MagicAnimate, and it's a prior work rather than a concurrent one. But this is a tiny point since ShowMaker has already demonstrated better visualization quality for gestures.

---

> > > ### Author Response · Authors · 2024-08-14
> > >
> > > We greatly appreciate your response and the valuable insights. In our revised manuscript, we will update the discussion and comparison of MagicPose in the related work and experiments sections, respectively.  In terms of out-of-domain data, we agree with your suggestion and we will make efforts to demonstrate its generalization ability on multiple styles of conversational videos.

---

### Official Review · Reviewer_R54L · 2024-07-07

**Soundness:** 3
**Presentation:** 3
**Contribution:** 3
**Rating:** 7
**Confidence:** 5

**Summary:**

This paper proposes a novel conversational human video generation framework named showmaker based on the fine-grained diffusion model. This paper proposes a novel Key Point-based Fine-grained Hand Modeling module and construct a key point-based codebook to handle the challenging hand generation. Meanwhile, this paper extracts facial texture features and global identity features from the aligned face and integrates them into the diffusion process for face enhancement. Sufficient experiments demonstrate the superiority of the proposed framework, and the generated human videos are of high quality.

I have read the comments of other reviewers and the response of the authors, I tend to give a accept score.

**Strengths:**

1.The paper is well written and clearly explains the motivations for the design as well as important technical details.
2.The human video generation framework introduced in this paper holds significant practical value, particularly in conversational scenarios like TV shows.
3.﻿The proposed Face Recapture module contributes to generate accurate face regions.
4.The paper has conducted sufficient experiments that demonstrate the superiority of the proposed model, and the generated videos are highly satisfactory.

**Weaknesses:**

1.Although the author compares the proposed method with talking head synthesis in the demo, it would be more beneficial to include this comparison in the main text or as an appendix, as it is crucial for understanding the contribution.
2.The presence of several writing errors in the paper impedes the correct comprehension of its content.
3.The details regarding some network structures and inference processes, such as the pose encoder, are insufficiently described and require further elaboration.

**Questions:**

1. The input for the Face Recapture Module in Figure 1 incorporates a reference pose, whereas the input in Figure 2 lacks such a reference pose.
2. Shouldn't line 140 on page 3 be replicated F times along the temporal dimension?
3. What is the rationale behind selecting a VAE encoder as the texture extractor in the Face Recapture Module?
4. Could you elaborate on the shapes of Gh and Gf? Additionally, why was a discrete codebook chosen to represent hand information? What advantages does this design offer?

**Limitations:**

Authors adequately discuss the limitations and potential negative societal impact

---

> ### Author Rebuttal · Authors · 2024-08-07
>
> # Response to Reviewer R54L
> Thanks very much for your valuable suggestions and for pointing out the typos in the manuscript. Here are my responses.
>
> - Q1. About the comparison with talking head synthesis.
>
>   A1. Thanks for your suggestion, we will add the talking head comparison in the demo to the main manuscript to reflect that our task is an extension of the talking head.
>
> - Q2. About the structure of the pose encoder.
>
>   A2. For a detailed description of the Pose Encoder, please refer to Figure 2 in the pdf.
>
> - Q3. About some typoes.(Response to Weakness 2 and Questions 1, 2)
>
>   A3. Sorry for the confusion. The "Reference Pose" in Figure 1 (d) should be removed. It should be repeated F times instead of L times in line 140 on page 3. We have corrected the known writing errors and will carefully revise the manuscript.
>
> - Q4. The reason for choosing the VAE encoder as the texture extractor in the Face Recapture Module?
>
>   A4. The VAE pre-trained on large-scale datasets has excellent compression capabilities and can capture local texture information of the face region well. Combined with global facial identity features, it can comprehensively represent facial information. Additional results are shown in Figure 3 in the pdf file, it can be seen that our method can generate satisfactory facial details due to the Face Recapture module.
>
> - Q5. About the shapes of $G_h$ and $G_f$. The advantages of adopting a discrete codebook chosen to represent hand information.
>
>   A5. Sorry for our oversight. The shape of $G_h$ is $B\times F\times 2\times 512$ and the shape of $G_f$ is $B\times F\times h\times w\times 768$. Representing hand information with a discrete codebook is robust. Moreover, the designed codebook consists of a set of orthogonal basis directions, which can fully represent hand structure and texture. (Please refer to A4 for Reviewer HRx2 and A3 for Reviewer jdWr for more motivation.)

---

### Official Review · Reviewer_HRx2 · 2024-07-08

**Soundness:** 3
**Presentation:** 2
**Contribution:** 3
**Rating:** 5
**Confidence:** 4

**Summary:**

This paper proposes a 2D human video generation framework called ShowMaker, which can generate half-body conversational videos based on 2D keypoints as motion conditions. ShowMaker includes a texture enhancement module for the face and hands, and demonstrates some effectiveness.

**Strengths:**

1. ShowMaker achieves good results in generating texture details, with improvements over the compared methods.
2. The motivation behind ShowMaker is straightforward, and the method design is simple.

**Weaknesses:**

1. The experimental settings lack clarity, especially regarding the balance of training duration for each ID in the testing examples. It is unclear how much training data corresponds to each ID in the visualized examples, and more details are needed.
2. The method proposed by the authors has some improvements in details compared to the similar method Make-an-Anchor. However, from Figure 4, it seems that the generation of facial details is unstable, resulting in some color changes (such as in the beard area), which raises concerns about the facial generation capability of the proposed method.
3. The ablation studies only excluded the face modeling and hand modeling modules. However, introducing region images to enhance the generation results is a straightforward approach in human body generation tasks and could serve as a baseline for comparison.

**Questions:**

1. In Table 1, it is unclear what "One ID" represents. It is also unclear whether the training set only contains data for the testing IDs. Additionally, it is unclear whether the compared methods, such as Animate Anyone and TPS, were also trained on the testing IDs. More details are needed to clarify these points.
2.  Are the IDs included in the training set and testing set are completely identical?

**Limitations:**

Please refer to the questions and weaknesses.

---

> ### Author Rebuttal · Authors · 2024-08-07
>
> # Response to Reviewer HRx2
> Thanks very much for taking time out of your busy schedule to review our manuscript.  We reply as follows.
>
> - Q1. The experimental settings. (Response to weakness 1 and question 2)
>
>   A1. Our goal is to extend the talking head task to generate an expressive half-body conversational speaker. Therefore our experimental setting is not a one-shot animation. Following the settings of Make-Your-Anchor, the IDs of the training set and the test set are consistent, and the test set and training set do not overlap.
>
>   &emsp; For detailed dataset information, please refer to Table 1 in the pdf file. Due to space limitations, we will place the additional experimental details in the appendix.
>
> - Q2. About "One ID" and Animate Anyone and TPS.
>
>   A2.  Since Make-Your-Anchor provides only one pre-trained model of Seth without the training code before our submission. We failed to reproduce its experiments on other subjects. Therefore, we compared our approach with it only on the Seth data ("One-ID") for fairness.
>
>   &emsp; Please also note that the training and test sets of Animate Anyone and TPS are consistent with ours.
>
> - Q3. About some color changes in the results.
>
>   A3. We suppose that the color changes mentioned by the reviewers may come from lighting changes in the facial region. Please refer to the videos in our supplementary videos and additional results in Figure 3 in the pdf file,  our results show temporally consistent texture in the facial region due to the Face Recapture module.
>
> - Q4. About introducing region images as a baseline for comparison.
>
>   A4. Thanks for your suggestion. Actually, we follow the idea of ​​using the regional image to enhance the results. In our Face Recapture Module, we use cropped and aligned face images to enhance the facial details in the generated results. However, when dealing with hand regions, there are two major problems:
>
>   &emsp; (1) Since they only occupy a small proportion of the image, it is difficult to capture enough textured details within the cropped hand images.
>
>   &emsp; (2) Hand poses, expressed in more degrees of freedom, are more complex when compared with head poses. The hand poses from the reference image and driving signals may be very different, which may alleviate the contribution of texture from the cropped hand image.
>
>   &emsp; Therefore, we map the driving hand key points to a high-dimensional space and adopt a discrete codebook for hand enhancement (refer to A3 for Reviewer jdWr).
>
>   &emsp; We still follow your suggestion to conduct an experiment and the results are illustrated in Figure 1 (the second row). Here, the cropped reference hand image is sent to the VAE encoder for feature extraction and then fed into the hand attention module. The results show that the hand modeling strategy we proposed enables the best synthesis of hand details and structures among all these variants.

---

> > ### Comment · Reviewer_HRx2 · 2024-08-13
> >
> > Thank you for the response, which has addressed the points I previously did not understand. I am inclined to maintain my rating.

---

> > > ### Author Response · Authors · 2024-08-13
> > >
> > > Thank you very much for your reply and insightful comments. Please let us know if you have any other questions or concerns, and we will respond to them carefully.

---

### Official Review · Reviewer_jdWr · 2024-07-16

**Soundness:** 4
**Presentation:** 3
**Contribution:** 3
**Rating:** 7
**Confidence:** 4

**Summary:**

This paper represents an early attempt to advance the field of 2D digital human synthesis in real-world scenarios, extending the traditional talking head task to include more complex body movements, such as hand gestures. The authors present a novel framework utilizing dual-stream diffusion models for full-body video synthesis, complemented by two specialized modules designed for fine-grained detail modeling. Specifically, the proposed Key Point-based Fine-grained Hand Modeling effectively addresses the challenge of hand gesture generation through the integration of a key point-based codebook and high-dimensional positional encoding. Additionally, the Face Recapture module enhances facial textures and identity consistency, enabling the generation of more realistic and vivid 2D avatar videos.

**Strengths:**

1) This paper outlines an exciting approach for digital human communities, focusing on the synthesis of 2D controllable human body videos with detailed modeling. Different from existing works like "Animate Anyone" that primarily focus on generalized human body movement generation across various subjects, this paper presents a promising direction for digital human communities by focusing on fine-grained modeling and controllable video synthesis of human body videos, especially hand gestures, which has the potential to achieve better human-computer interaction with our body gesture and benefit our demands in real-world scenarios.

2) The paper is well-written and easy to follow. The motivation for the whole task and the two additional modules is clear. The supplementary video provides robust results and convincible comparison with sota methods, which further demonstrate the effectiveness of all the designs proposed in this paper.

**Weaknesses:**

There are few mistakes and typos in the paper:
1) The Face Recaption module takes “Reference Pose” as input in Fig 1 (d), which is not mentioned in Fig 2 and Sec 3.3. The description of Face Recaption module should be the same throughout the full manuscript.

2) Eq (5) and Eq(6) are nearly the same. The authors can merge them into one equation.

**Questions:**

1) I understand the authors adopted the positional encoding to enhance the information provided by the sparse hand key points, but I’d like to see a further analysis of the positional encoding used in the Hand Modeling module. Since it cooperates with the key point-based codebook, I am wondering if a well-trained codebook is already enough for hand restoration.

2) Have the authors adopted any alignment strategy in cross-driving experiments? In my understanding, cross-driving animation usually suffer from mis-alignment due to the subject differences.

**Limitations:**

The authors adequately addressed the limitations and negative societal impact.

---

> ### Author Rebuttal · Authors · 2024-08-07
>
> # Response to Reviewer jdWr
> Thanks very much for your careful review comments. We'll answer your questions one by one.
> - Q1: About Fig 1 (d).
>
>   A1. Thanks for pointing out this mistake. We will remove the “Reference Pose” from Figure 1 (d) and refine the whole figure in the revised manuscript.
>
> - Q2: About Eq (5) and Eq (6).
>
>   A2. Thanks for your suggestion. We will replace Eq (5) and Eq (6) with a unified definition of the attention operation and describe the different inputs used in Eq (5) and Eq (6).
>
> - Q3. The effect of position encoding on generation results.
>
>   A3. Please refer to Figure 1 (the third row) in the pdf file, the comparison results demonstrate that position encoding brings a significant performance improvement in both textural and structural details of human hands. The key points of the hands are distributed in a low-dimensional space and are relatively clustered, directly using the coordinates of the hand key points as inputs of the Hand Modeling module cannot well capture the structure and texture information from the hand.
>
> - Q4. Alignment strategy in cross-driving experiments.
>
>   A4. For the cross-driving setting, we adopt a pose alignment strategy that roughly aligns the driving signals from other subjects with the ones of the target person. The key points of our pose alignment strategy are as follows:
>
>     &emsp; (1) When preparing the training data, we first produce the DWPose results from each frame and set the center of shoulders as the cropping center. Then we crop the original video frame by using an adaptive cropping size, where the cropping size is designed as a fixed ratio of the shoulder width. This operation forces the human body to lie in a roughly consistent position.
>
>     &emsp; (2) In the inference stage, we further try to bridge the gap between body shapes by scaling the driving poses to match the reference pose. The scaling ratios of width and height are defined as $w_r/w_d$, $h_r/h_d$. Here, $w$ represents the shoulder width, and $h$ represents the height of the human body. We will involve a detailed description of $w$ and $h$ in the appendix.
>
>    &emsp; The above strategy enables satisfactory pose alignment between different subjects under our scenario.

---

> > ### Comment · Reviewer_jdWr · 2024-08-13
> >
> > My concerns about the the alignment strategy and the effect of position encoding have been well addressed in the rebuttal. I am willing to increase my score.

---

> > > ### Author Response · Authors · 2024-08-14
> > >
> > > Thank you very much for your response, we are glad to address your concerns!

---

### Author Rebuttal · Authors · 2024-08-07

# To all reviewers and ACs
Thanks very much for all the reviewers' efforts and suggestions. We appreciate the positive comments on the following:
1. The generation framework holds significant practical value, particularly in conversational scenarios like TV shows.
2. The paper is well-written and clearly explains the important technical details.
3. The motivation behind ShowMaker is straightforward and the designed model is clear and convincible.
4. The proposed model achieves good results in generating texture details, with improvements over the compared methods, and the demo presents satisfying hand and face modeling results.

We believe the remaining issues can be fully addressed. We will respond to your comments one by one below.
Some relevant figures and tables are provided in the pdf file.

---

### Decision · Program_Chairs · 2024-09-25

**Decision:**

Accept (poster)

**Comment:**

Reviewers mostly agree that the proposed system is well-designed, and provides good results for the less studied problem of half body generation. The authors' rebuttal addressed most of the reviewers' concerns who in many cases increased their score. Overall there seems to be consensus for acceptance.